# The ultrastructural characteristics of bile canaliculus in porcine liver donated after cardiac death and machine perfusion preservation

Yo Ishihara[1☺¤], Hiroki Bochimoto[1,2☺]*, Daisuke Kondoh[3], Hiromichi Obara[4], Naoto Matsuno[1,5]

**1** Department of Transplantation Technology and Therapeutic Development, Asahikawa Medical University, Asahikawa, Japan, **2** Division of Aerospace Medicine, Department of Cell Physiology, The Jikei University School of Medicine, Minato-ku, Japan, **3** Laboratory of Veterinary Anatomy, Obihiro University of Agriculture and Veterinary Medicine, Obihiro, Japan, **4** Department of Mechanical Engineering, Tokyo Metropolitan University, Hachioji, Japan, **5** Department of Surgery, Asahikawa Medical University, Asahikawa, Japan

☺ These authors contributed equally to this work.
¤ Current address: Shonan Kamakura General Hospital, Kamakura, Japan
* botimoto@jikei.ac.jp

**Data Availability Statement:** All relevant data are within the paper.

**Funding:** This work was supported by the Grants-in-Aid for Scientific Research (C) (KAKENHI)

## Abstract

The effects of each type of machine perfusion preservation (MP) of liver grafts donated after cardiac death on the bile canaliculi of hepatocytes remain unclear. We analyzed the intracellular three-dimensional ultrastructure of the bile canaliculi and hepatocyte endomembrane systems in porcine liver grafts after warm ischemia followed by successive MP with modified University of Wisconsin gluconate solution. Transmission and osmium-maceration scanning electron microscopy revealed that lumen volume of the bile canaliculi decreased after warm ischemia. In liver grafts preserved by hypothermic MP condition, bile canaliculi tended to recover in terms of lumen volume, while their microvilli regressed. In contrast, midthermic MP condition preserved the functional form of the microvilli of the bile canaliculi. Machine perfusion preservation potentially restored the bile canaliculus lumen and alleviated the cessation of cellular endocrine processes due to warm ischemia. In addition, midthermic MP condition prevented the retraction of the microvilli of bile canaliculi, suggesting further mitigation of the damage of the bile canaliculi.

## Introduction

The shortage of brain dead donors for liver transplantation is a serious problem worldwide [1]. Although donors with circulatory arrest have the potential to expand the transplanted liver pool [2,3], post-circulatory arrest liver grafts induce high rates of primary nonfunction and ischemia-reperfusion injury after transplantation [4]. In particular, a high risk of acute and chronic rejection, including ischemic bile duct damage, and biliary complications has been reported [5]; thus, the development of liver graft preservation methods after circulatory arrest is required to overcome these problems [1].

No.17K10503 of Japan Society for the Promotion of Science to N.M. URL of the funder's website: http://www.jsps.go.jp/english/. The funders had no role in study design, data collection and analysis, decision to publish, or preparation of the manuscript. In addition, all of the authors received no salary from any funder of this research.

**Competing interests:** The authors have declared that no competing interests exist.

Existing cold storage organ preservation techniques fail to preserve marginal donor grafts [6]. On the other hand, machine perfusion (MP) of post-circulatory arrest donor liver grafts has been reported to have numerous advantages [6–25], and the optimal conditions of MP, including perfusion temperature, oxygenation status, flow rate, steady flow and pulsatile flow, have been discussed [26–33]. Recently, hypothermic MP (HMP) has been established to maintain the functions of liver grafts, and its application in clinical practice has begun [34–46]. On the other hand, warm perfusion has also been reported as an advanced MP method that maintains the liver graft functions [2,38,47–56]. Warm perfusion had introduced to maintain liver grafts at a more physiologic temperature compared with HMP to offers the opportunity to assess and possibly repair a metabolically active liver graft. Our previous reports indicated that the midthermic MP (MMP), one type of warm perfusion [57], reduces the hepatocellular enzyme release [58,59]. In addition, we confirmed that hepatocytes of DCD liver grafts after MMP retain a functional ultrastructure compared to HMP, by using the observation method of scanning electron microscopy after osmium-maceration (OM-SEM) [60,61]; ultrastructural characteristics of hepatocytes are reported to reflect the function of the transplanted liver [60].

One of the important physiological functions of hepatocytes is the production and secretion of bile [62]. For liver grafts, MP has the potential to not only inhibit the development of post-transplant biliary complications, including ischemic cholangiopathy [17,63–67], but also to protect the bile canaliculus [68]. We evaluated the ultrastructural changes in the bile canaliculi and hepatocytes around them at four hours after HMP or MMP using OM-SEM and transmission electron microscopy (TEM). As a result, the bile canaliculi that regressed one hour after warm ischemia showed a strong tendency to recover after MP, especially in MMP, suggesting the preventative effects of HMP and MMP on bile canaliculi-related functions in liver grafts.

## Materials and methods

### Animals

We purchased domestic female pigs (cross-bred Large White, Landrace, and Duroc pigs; age, 2–3 months; body weight, approximately 25 kg) from Taisetsusanroku-sya Co., Ltd. (Asahikawa, Japan). The pigs were kept in a well-ventilated room with a 12-h light: dark cycle, controlled temperature and humidity, and *ad libitum* access to food and water. All experiments were performed according to the Guide for the Care and Use of Laboratory Animals at Asahikawa Medical University, and the procedures were approved by the Institutional Animal Ethics Committee of the Clinical Research Center, Asahikawa Medical University (permit no. 14172).

### Machine perfusion preservation

Livers harvested from pigs were connected and perfused with a MP system (Fig 1), as described previously [61]. The system was composed of two separate circulating perfusion circuits, which had a roller pump, for the hepatic artery (HA) and portal vein (PV). Each circuit had a flow meter and a pressure sensor, allowing pulsatile and non-pulsatile flow, respectively. Additionally, an oxygenator was installed in the the upstream of the circuits for the PV and HA were connected via plastic connectors to each of the hepatic vessels. The MP systems had waterproof thermocouples that measured the solution and organ temperatures, and a dissolved oxygen meter. The flow conditions and temperatures of the preservation solution were recorded by a system-installed computer. In the systems, the organ chamber temperature was controlled by ice-cold water and a heat exchanger. The flow rate was mainly set to 0.22 mL/min/g for the PV and 0.06 mL/min/g for the HA, as described previously [61].

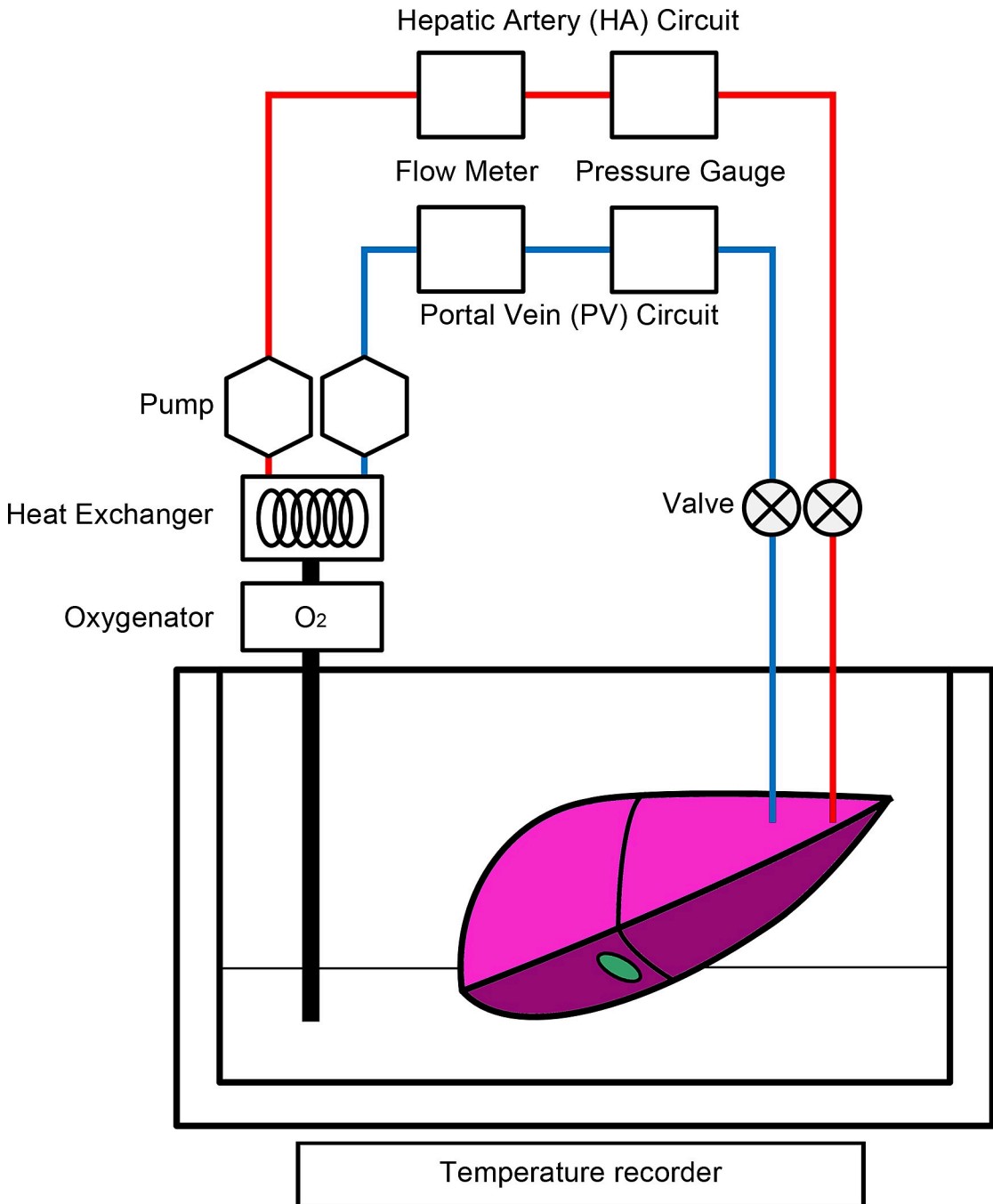

**Fig 1. Schematic representation of the continuous machine perfusion system.**

### Preparation and preservation of the liver donated after cardiac death

Pigs were used as liver graft donors. These pigs were intubated and ventilated with inhalation anesthesia by isoflurane (Forane; Abbott, Japan), and laparotomized. Immediately after laparotomy, liver tissue samples from the liver surface were obtained by biopsy as a control. Then the pigs were intravenously injected with potassium chloride to induce circulatory arrest followed by the withdrawal of ventilation, as described previously [61]. The time point of the

induction of circulatory arrest was defined as 0 minutes of warm ischemia. During warm ischemia, hepatic artery and portal vein were isolated to connect with organ flush lines, and at 60 minutes of warm ischemia, the liver tissue samples were obtained from distinct regions of the liver surface. Immediately after tissue sampling, the liver grafts were harvested and subsequently flushed with Euro-Collins solution via the HA and PV routes at 8°C on the back table. After the initial flushing, the flush routes were connected to the perfusion preservation machine and the liver grafts were continuously perfused for four hours with modified University of Wisconsin gluconate solution (sodium gluconate 17.5 g, KH2PO4 3.4 g, trehalose 10 g, glutathione 0.9 g, adenosine 1.3 g, HEPES 4.7 g, penicilline 200,000 U, dexamethasone 16 mg, MgSO4 1 g, caffeine 4 g, polyethylene glycol 10 g, and glycine 1 g per 1 L). The liver grafts were conserved at a constant temperature of 8°C as HMP (n = 4) or gradually warmed from 8°C to 22°C during perfusion as MMP (n = 5). After MP, the liver tissue samples were biopsied from the well-perfused region of graft surface in each group. Liver sample blocks were immediately fixed with an appropriate fixative for the analysis, as described below. The degree of biliary injury at four hours after MP were evaluated based on the alkaline phosphatase level in perfusate collected from the suprahepatic vena cava of liver grafts in each MP group, as described previously [69]. These alkaline phosphatase data were presented as the mean ± SEM, and unpaired two-tailed $t$-tests were used to compare the significance of differences between groups A and B.

## Transmission electron microscopy

The liver tissue samples were trimmed into small blocks and fixed with 2% glutaraldehyde and 2% paraformaldehyde in 0.1 M phosphate buffer (PB) for two hours at 4°C. After fixation, the blocks were washed 3 times with PB containing 7.5% sucrose and post-fixed with 1% osmium tetroxide ($OsO_4$) in PB for two hours at 4°C. After washing thoroughly with PB containing 7.5% sucrose, the blocks were dehydrated with a graded series of ethanol. After dehydration, the samples were transferred in propylene oxide, infiltrated and then embedded in epoxy resin (Epon 812). Ultrathin section (80 nm thick) were cut, stained with uranyl acetate and lead citrate, and observed using an HT7700 transmission electron microscope (Hitachi High Technologies, Tokyo, Japan).

## Osmium-maceration for SEM

For SEM observation, the osmium maceration method was applied to the liver tissue samples, as described previously [61]. In brief, liver samples cut into small pieces were fixed with 0.5% glutaraldehyde and 0.5% paraformaldehyde in PB for 30 min at 4°C. After fixation, the liver blocks were directly immersed in 1% $OsO_4$ in PB for four hours at 4°C. The samples were then washed thoroughly with PB and transferred into dimethyl sulfoxide solution in order of 25 to 50% each for 30 min for cryoprotection. The samples were then frozen on a deeply chilled aluminum metal plate with liquid nitrogen, and cracked into two particles with a screwdriver and a hammer. After freeze cracking, the samples were thawed in 50% dimethyl sulfoxide solution, washed thoroughly in PB and transferred into 0.1% diluted $OsO_4$ in PB for 96 hours at around 20°C under light for maceration. After the maceration period, the samples were immersed into 1% $OsO_4$ in PB for one hour for post-fixation and then thoroughly washed with PB. The samples were transferred into 1% tannic acid in PB for two hours, subsequently washed with PB, and then immersed in 1% $OsO_4$ in PB for one hour for conductive staining. The liver samples were dehydrated in graded series of ethanol and immersed in tert-butyl alcohol. After freezing, the samples were lyophilized in an ES2030 freeze-dryer (Hitachi Koki Co., Ltd., Tokyo, Japan). The dried specimens were then mounted onto a metal plate and lightly coated

with platinum-palladium in an E1010 ion sputtering device (Hitachi Koki). These finally processed specimens were observed in secondary electron-mode by a field emission S4100 scanning electron microscope (Hitachi High Technologies).

## Results

### Ultrastructure of the normal bile canaliculi observed by OM-SEM and TEM

First, we established the overall shape of bile canaliculus by OM-SEM and TEM. In control livers, OM-SEM revealed that hepatocytes mutually formed the bile canaliculi with microvilli between the plasma membranes of contiguous hepatocytes (Fig 2A and 2B). The bile canaliculi were often accompanied by several small stacks of Golgi apparatus around the cytoplasm of the constituent hepatocytes (Fig 2B). The small vacant spaces without any other endomembrane organelles around the bile canaliculi were often found (Fig 2B). The corresponding findings were also obtained by TEM observation (Fig 2C). The vacant space around the bile canaliculi observed by OM-SEM corresponded to the cytoplasmic region without endomembrane organelles (Fig 2C). These findings showed that the details and three-dimensional conformation of the bile canaliculi and the related intracellular components could be visualized by OM-SEM with complementary TEM observation.

### Changes in the ultrastructure of the bile canaliculi after warm ischemia

The continuous hypoxic exposure of liver grafts induced by one hour of warm ischemia caused the cessation of bile production and morphological abnormalities of the bile canaliculi. After warm ischemia, OM-SEM revealed the large vacuoles in hepatocytes (Fig 3A, colored red), as described previously. Although the bile canaliculi seemed normal at low magnification (Fig 3A), the cross-sectional area of the lumen of the bile canaliculi after warm ischemia tended to become smaller in comparison to controls (Figs 2B and 3B). In contrast with the controls, small stacks of Golgi apparatus were rarely detected around these bile canaliculi (Figs 2B and 3B). The small vacant spaces around the bile canaliculi were observed similarly to controls, and these space corresponded to the cytoplasmic area without any endomembrane organelles observed by TEM (Fig 3B and 3C). These findings showed that warm ischemia causes ultrastructural destruction of bile canaliculi and the related intracellular subsets, reflecting the decreased bile production induced by hypoxic exposure.

### Recovery of the ultrastructure of bile canaliculi by HMP and MMP

Even after liver graft preservation with four hours of HMP or MMP, almost no bile was collected from the bile ducts of the liver grafts. However, ultrastructural restoration of the bile canaliculi was found to have occurred, particularly after MMP.

After four hours of HMP, OM-SEM revealed swollen mitochondria in many hepatocytes (Fig 4A). The cross-sectional area of the lumen of the bile canaliculi after HMP was restored from the changes that were observed in warm ischemia (Figs 3B and 4B); however, the restoration was decreased in comparison to that in the controls (Figs 2B and 4B). This restoration of the bile canaliculi was more clearly indicated by TEM (Fig 4C), whereas the microvilli in the bile canaliculi tended to regress after HMP (Fig 4C).

OM-SEM revealed that the hepatocytes after MMP included macro-autophagosomes and functional forms of mitochondria (Fig 5A), indicating that MMP was more protective for hepatocytes than HMP, as described previously. Additionally, after MMP, the cross-sectional area of the lumen of the bile canaliculi almost recovered to the control level (Fig 5B and 5C), and the microvilli in the bile canaliculi were also maintained (Fig 5B and 5C).

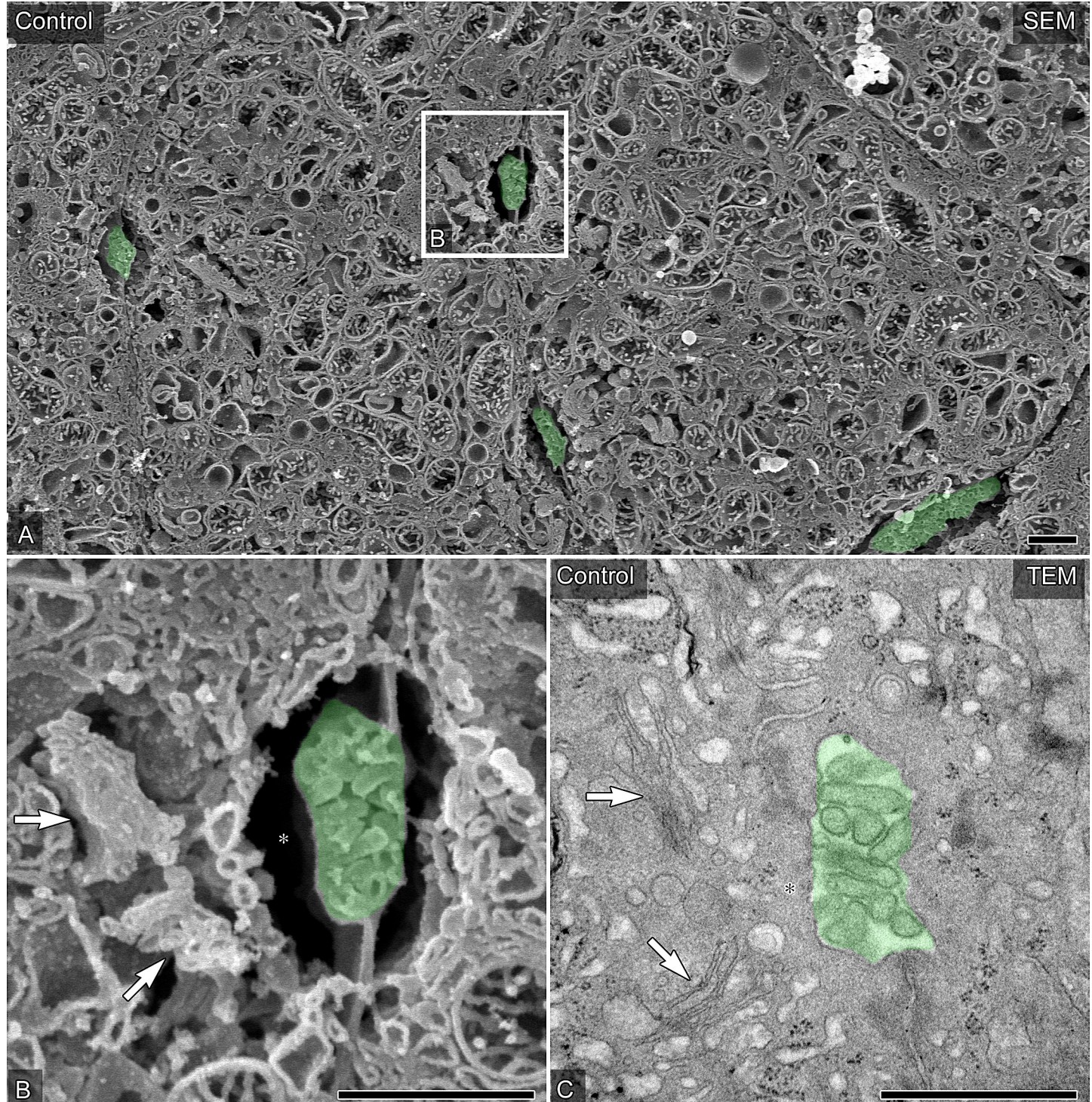

**Fig 2. The ultrastructure of the bile canaliculi in porcine hepatocytes of the control liver.** (A and B) Representative hepatocytes and bile canaliculi were observed by SEM in osmium-macerated control porcine liver graft samples. The partial area indicated in A was further photographed at a higher magnification (B). (C) Typical bile canaliculi were identified in the ultrathin sections of the Epon 812-embedded control liver tissue. Bile canaliculi are colored green. Arrows indicate the Golgi apparatus, and asterisks indicate vacant spaces without any other endomembrane organelles around the bile canaliculi. Bars = 1 μm.

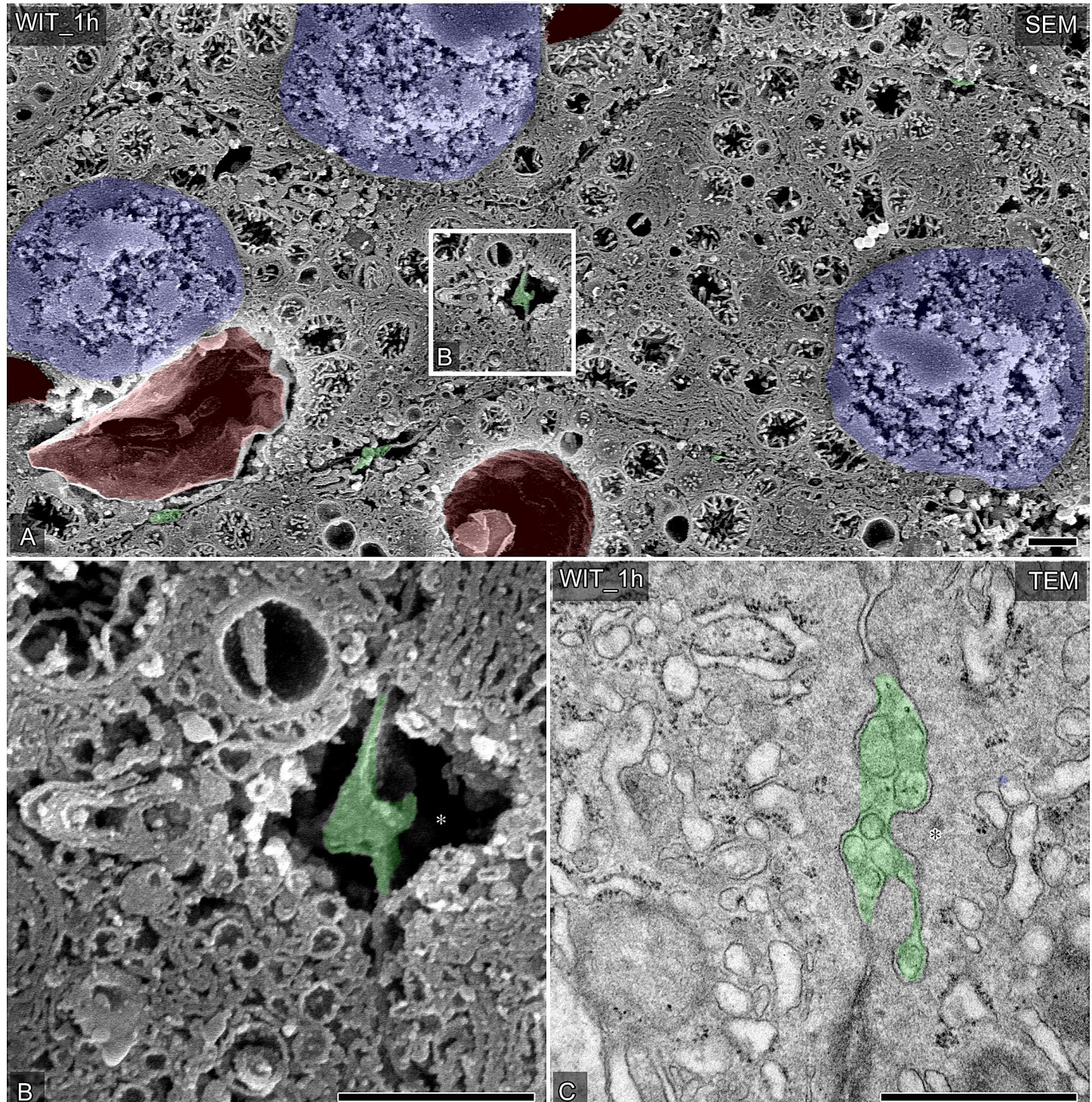

**Fig 3. Changes in the ultrastructure of the bile canaliculi in porcine hepatocytes after warm ischemia.** (A and B) Representative hepatocytes and bile canaliculi were observed by SEM in osmium-macerated porcine liver graft samples after warm ischemia for 60 minutes. The partial area indicated in A was further photographed at a higher magnification (B). Bile canaliculi are colored green, nuclei are colored blue, huge vacuoles are colored red. Asterisks indicate vacant space without any other endomembrane organelles around the bile canaliculi. (C) Typical bile canaliculi were identified in the ultrathin sections of Epon 812-embedded tissues from liver graft samples after warm ischemia for 60 minutes. Bile canaliculi are colored green and asterisks indicate the vacant space without any other endomembrane organelles around the bile canaliculi. Bars = 1 μm.

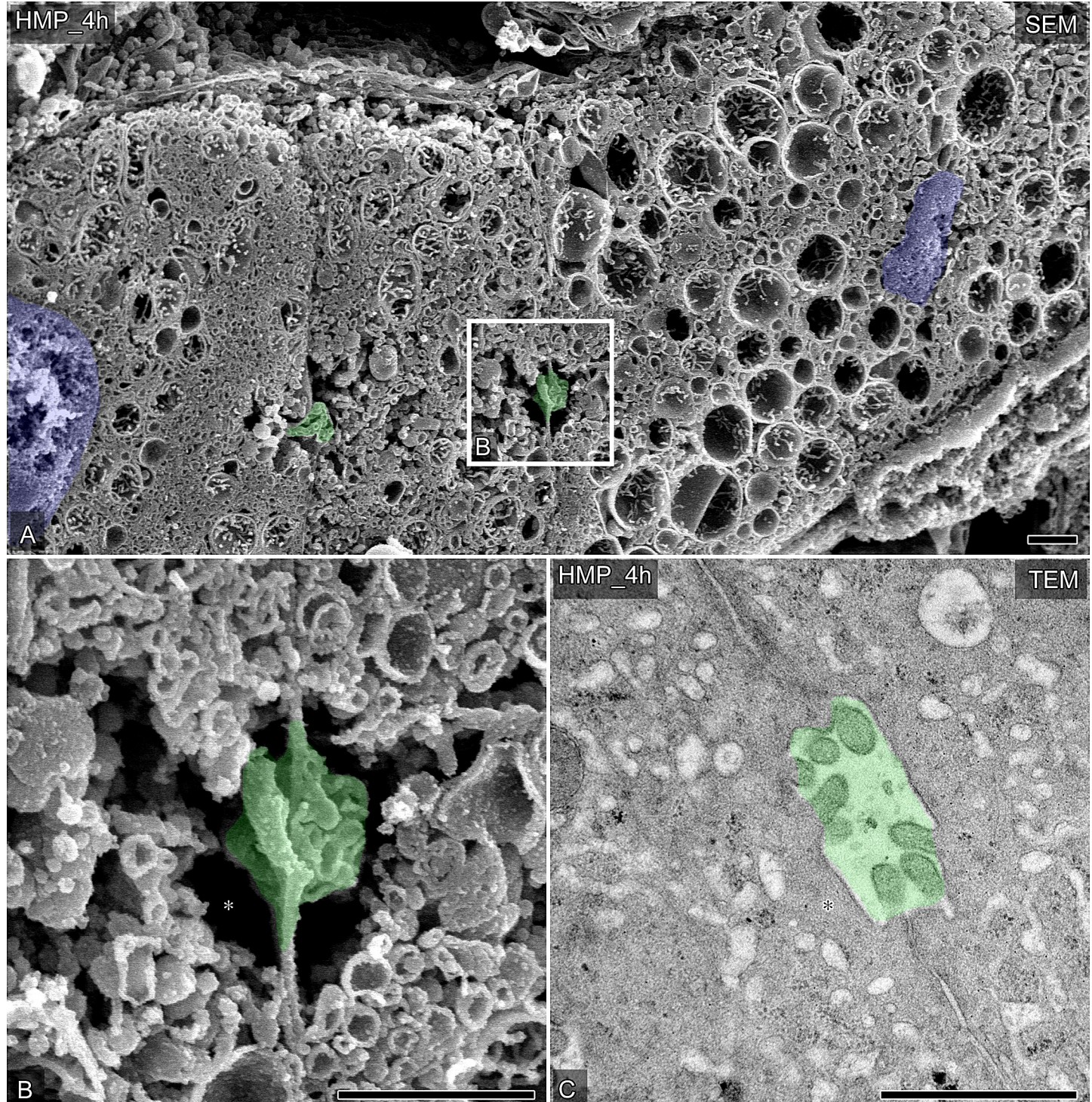

**Fig 4. The ultrastructural changes of the bile canaliculi in porcine liver grafts preserved by HMP.** (A and B) Representative hepatocytes and bile canaliculi were observed by SEM in osmium-macerated porcine liver graft samples preserved by HMP for 4 h after 60 minutes of warm ischemia. The partial area indicated in A was further photographed under higher magnification (B). Bile canaliculi are colored green, nuclei are colored blue. Asterisks indicate vacant space without any other endomembrane organelles around the bile canaliculi. (C) Typical bile canaliculi were identified in the ultrathin sections of Epon 812-embedded tissues from liver graft samples preserved by HMP for 4 h after 60 minutes of warm ischemia. Bile canaliculi are colored green. Asterisks indicate the vacant space without any other endomembrane organelles around the bile canaliculi. Bars = 1 μm.

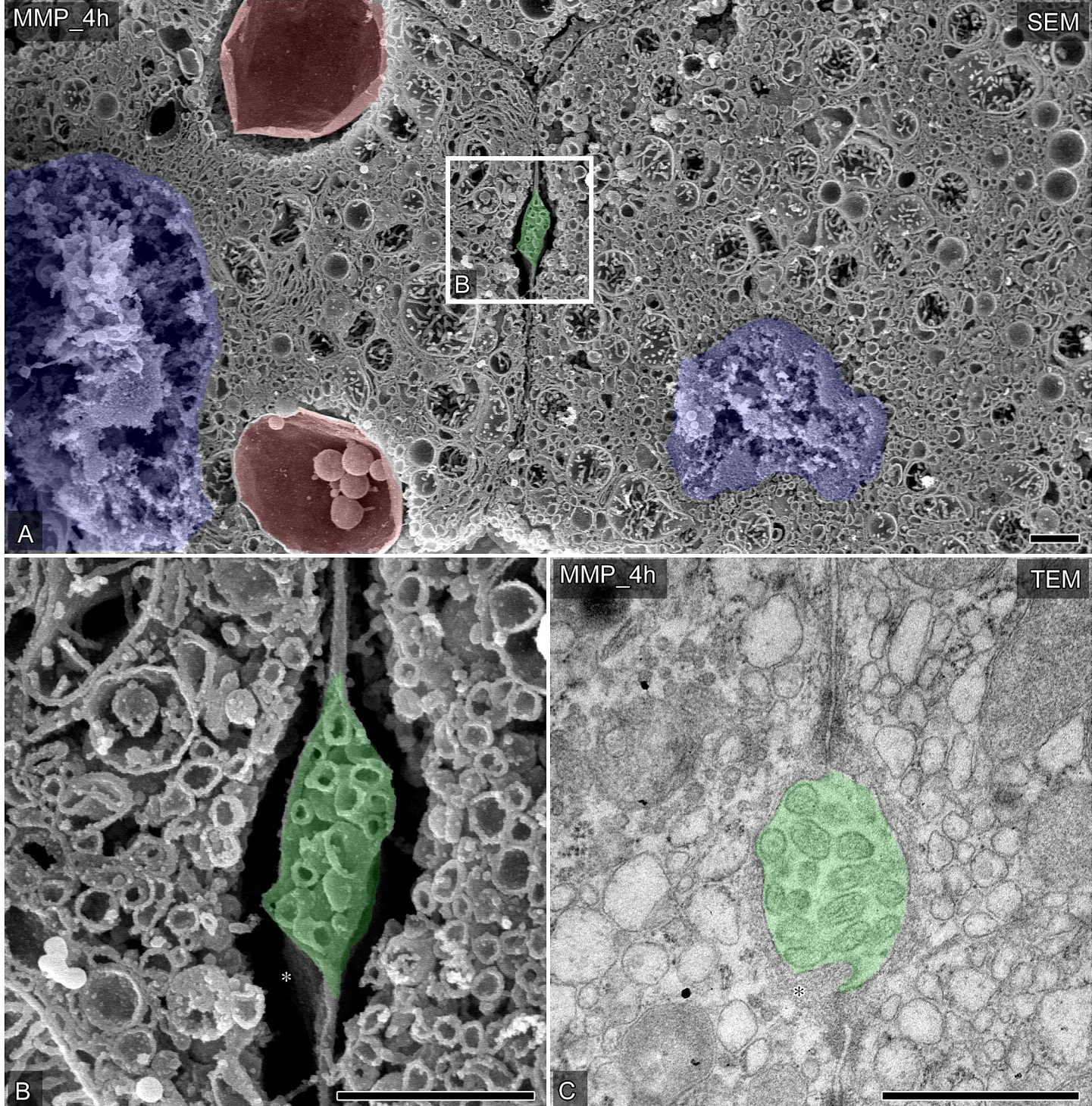

**Fig 5. The ultrastructural changes of the bile canaliculi in porcine liver grafts preserved by MMP.** (A and B) Representative hepatocytes and bile canaliculi were observed by SEM in osmium-macerated porcine liver graft samples preserved by MMP for 4 h after 60 minutes of warm ischemia. The partial area indicated in A was further photographed under higher magnification (B). Bile canaliculi are colored green, nuclei are colored blue, huge vacuoles are colored red. Asterisks indicate vacant space without any other endomembrane organelles around the bile canaliculi. (C) Typical bile canaliculi were identified in the ultrathin sections of the Epon 812-embedded tissues from liver graft samples preserved by MMP for 4 h after 60 minutes of warm ischemia. Bile canaliculi are colored green. Asterisks indicate vacant space without any other endomembrane organelles around the bile canaliculi. Bars = 1 μm.

Correspondingly, the value of alkaline phosphatase in perfusate after four hours of MMP (18.0 ±3.1 IU/L) was significantly lower in comparison to HMP (26.3±1.0 IU/L) (S1 Fig), indicating that MMP suppressed the increased biliary enzyme release in comparison to the HMP. Nevertheless, after preservation with both HMP and MMP, small stacks of Golgi apparatus were rarely detected around the bile canaliculi (Figs 4B and 5B). These findings indicated that, MP preservation, especially MMP, of liver grafts after warm ischemia enabled the ultrastructure of the bile canaliculi to be maintained and restored.

## Discussion

In the present study, the porcine bile canaliculi after warm ischemia and after HMP and MMP preservation were analyzed by OM-SEM and complementary TEM methods. Okouhchi et al. [70] analyzed the ultrastructure of the rat liver after HMP preservation by OM-SEM, but did not describe the bile canaliculi. Our previous study using OM-SEM [61] described the bile canaliculi after MMP preservation under low magnification, but the detailed ultrastructure of the bile canaliculi was not established at that stage. The present study revealed the detailed ultrastructural changes of the bile canaliculi after both HMP and MMP preservation.

Morphological abnormalities of the bile canaliculi after warm ischemia were alleviated by subsequent MP, and this preventative effect was greater in MMP than in HMP. These findings were consistent with previous physiological reports [2,38,47–56], which suggested that the ultrastructure of the bile canaliculi is important from both morphological and functional perspectives.

One hour of warm ischemia was associated with regression of the bile canaliculus lumen; this change is consistent with our previous findings [61]. Moussa et al. [71] also reported the regression and disappearance of the bile canaliculi during warm ischemia based on observation by TEM. These findings may reflect autophagic changes in hepatocytes after warm ischemia suppressed the intracellular trafficking pathway [72]. In addition, small stacks of Golgi apparatus around the bile canaliculi were not found after warm ischemia. This distorted arrangement of Golgi apparatus was probably caused by hypoxia because the hepatocellular polarity is maintained by mitochondrial energy [73,74]. On the other hand, the organelle-poor cytoplasmic region of hepatocytes around the bile canaliculi and functional microvilli in the bile canaliculi was retained, even after warm ischemia. The accumulation of actin, a cytoplasmic component concentrated in the peri-biliary region and microvilli, occurs in hepatocyte cytoplasm around the bile canaliculi [75], and the actin in this region is disrupted under reperfusion rather than during warm ischemia [76], although the microtubules in hepatocytes are distorted after warm ischemia [77]. The present study supported the opinion that warm ischemia does not largely affect the cytoplasmic components around the bile canaliculi.

The cross-sectional area of the bile canaliculus lumen in liver grafts that regressed from warm ischemia tended to recover after both HMP and MMP preservation, suggesting the resumption of the secretion of bile canaliculi contents, namely bile salts. The secretion of bile salts from hepatocytes into the bile canaliculi is an important factor for activating bile production [78]. HMP and MMP did not seem to irregularly increase the bile duct pressure, although irregular peri-biliary actin accumulation and biliary dilatation caused by the ischemia-reperfusion treatment—reflecting increased bile duct pressure—has been reported [79]. These findings suggested that the initial cold perfusion phase present in both MMP and HMP may prevent mitochondrial damage in hepatocytes and lead to the resumption of bile salt secretion [65].

The bile canaliculus microvilli after HMP preservation tended to regress more in comparison to after MMP. The ultrastructure of the microvilli in the bile canaliculi is associated with

the bile secretion function [73,80,81], and ultrastructural changes of bile canaliculus microvilli during ischemia-reperfusion injury, drug injury, and cholestasis are characterized by microvilli withdrawal [81–87]. Retraction of the microvilli occurs due to damage of the cell membrane forming the bile canaliculi by protonated hydrophobic bile salts [65,88]. Since HMP consumes less oxygen than warm perfusion [89], hepatocytes may reduce ATP production due to decreased metabolism during HMP preservation. ATP depletion in hepatocytes alters transporter functions, such as the bile salt export pump and disrupts the balance between bile salts and phospholipids [65]. It is therefore considered that the neutralization of salts by phospholipids is weakened and that the cell membrane of the bile canaliculi may be more damaged during HMP than during MMP. Actually, it was also confirmed that the level of ALP, a marker of capillary bile duct damage, was lower in MMP than in HMP [90].

The present study was associated with several limitations. First, this study did not confirm the ultrastructural changes in the bile canaliculi in the liver transplanted or monitored after perfusion storage. Like a previous study of allogeneic liver transplantation of porcine liver grafts under conditions that were similar to the conditions in this study [91], the present MMP results are still in the preclinical stage. In addition, this study did not evaluate the effect in reperfusion at normothermia Thus, the present results should be investigated by further studies using liver transplant models that are clinically suitable for transplantation and reperfusion at normothermia ex situ.

Second, among the components of the biliary system, bile duct cells are very sensitive to ischemia [4]; however, this study did not examine the bile ducts. Future studies are needed to examine the ultrastructure of the bile duct in order to investigate the optimal conditions for clinical transplantation [65].

In conclusion, MP preservation alleviated the cessation of intracellular trafficking processes of hepatocytes caused by warm ischemia and restored the retracted bile canaliculus lumen. In addition, MMP temperature conditions prevented the retraction of the microvilli in the bile canaliculi by mitigating the damage to the cell membrane forming the bile canaliculi. In future study, more clinically appropriate MP conditions to preserve the functions of the liver grafts should be established by using normothermically reperfusion systems. To achieve the above objectives, further physiological studies are required to reveal the ultrastructural changes in liver constituent cells under various conditions, including different temperatures and different levels of oxygenation, during perfusion storage.

## Supporting information

**S1 Fig. Changes in the perfusate enzymes after warm ischemia and subsequent preservation by HMP or MMP.** The levels of alkaline phosphatase (ALP) in the perfusate at 4 hours after hypothermic and midthermic machine perfusion preservation. Data are shown as the mean ± SEM. Unpaired two-tailed $t$-tests were used ($p < 0.05$).
(TIF)

## Acknowledgments

We thank all of the lab members and colleagues for their helpful suggestions and assistance in the experiments. We are extremely grateful to Mr. Yoshiyasu Satake for carrying out all of the research.

## Author Contributions

**Conceptualization:** Hiroki Bochimoto, Naoto Matsuno.

**Funding acquisition:** Naoto Matsuno.

**Investigation:** Yo Ishihara, Hiroki Bochimoto, Daisuke Kondoh, Hiromichi Obara, Naoto Matsuno.

**Resources:** Yo Ishihara, Hiroki Bochimoto, Daisuke Kondoh, Hiromichi Obara, Naoto Matsuno.

**Supervision:** Naoto Matsuno.

**Writing – original draft:** Hiroki Bochimoto.

**Writing – review & editing:** Yo Ishihara, Hiroki Bochimoto, Daisuke Kondoh, Hiromichi Obara, Naoto Matsuno.

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
