## [Decision Letter · Decision Letter 0]

24 Feb 2020

PONE-D-19-34589

The ultrastructural characteristics of bile canaliculus in porcine liver donated after cardiac death and machine perfusion preservation

PLOS ONE

Dear Dr. Bochimoto,

Thank you for submitting your manuscript to PLOS ONE. After careful consideration, we feel that it has merit but does not fully meet PLOS ONE’s publication criteria as it currently stands. Therefore, we invite you to submit a revised version of the manuscript that addresses the points raised during the review process.

Please check and review all references cited in the manuscript to ensure they directly support the points made in the text. Please also ensure that the use of the English language is checked and edited as appropriate.

We would appreciate receiving your revised manuscript by Apr 09 2020 11:59PM. To enhance the reproducibility of your results, we recommend that if applicable you deposit your laboratory protocols in protocols.io, where a protocol can be assigned its own identifier (DOI) such that it can be cited independently in the future. For instructions see: http://journals.plos.org/plosone/s/submission-guidelines#loc-laboratory-protocols

We look forward to receiving your revised manuscript.

Kind regards,

Kourosh Saeb Parsy

Academic Editor

PLOS ONE

Journal Requirements:

a)    Please provide an amended Funding Statement that declares *all* the funding or sources of support received during this specific study (whether external or internal to your organization) as detailed online in our guide for authors at http://journals.plos.org/plosone/s/submit-now.  

b)    Please state what role the funders took in the study.  If any authors received a salary from any of your funders, please state which authors and which funder. If the funders had no role, please state: "The funders had no role in study design, data collection and analysis, decision to publish, or preparation of the manuscript."

Reviewers' comments:

Reviewer's Responses to Questions

**Comments to the Author**

1. Is the manuscript technically sound, and do the data support the conclusions?

Reviewer #1: Partly

Reviewer #2: Yes

2. Has the statistical analysis been performed appropriately and rigorously? 

Reviewer #1: N/A

Reviewer #2: N/A

3. Have the authors made all data underlying the findings in their manuscript fully available?

Reviewer #1: Yes

Reviewer #2: Yes

4. Is the manuscript presented in an intelligible fashion and written in standard English?

Reviewer #1: No

Reviewer #2: No

5. Review Comments to the Author

Reviewer #1: Thank you for asking me to review this paper by Ishihara et al. I am afraid I have a number of issues with the paper and experimental conclusions. In addition, or possibly explaining a number of my concerns, is a problem with the use of the English language.

1. Abstract and throughout paper: “Warm MP.” The authors use this term but it is not an accepted term in the field. What they are actually describing is controlled oxygenated reperfusion, as described by Thomas Minor’s group; the temperature does not go above 22°C.

2. Introduction. References

The authors include many references in the paper, but many are completely unrelated to the point they are trying to make. So, for example, references 4 and 5 do not discuss PNF and IRI post transplant; references 6 and 7 include one pig study and one NMP study of 30 patients compared to an historical control which do not address chronic rejection. A reference needs to be pertinent to the point being made, or it should not be there. I have not examined the other references in detail but I imagine similar issues pertain.

3. Introduction. “Recently, oxygenated cryogenic mechanical…”

Cryogenic is not the correct word – I suspect they mean hypothermic

4. Introduction. “The WMP method more closely approximates physiological conditions by gradually increasing the perfusion temperature from the initial low temperature range”

There is nothing physiological in humans or pigs about being rewarmed from 8°C.

5. Introduction “… deviating enzyme .”

What is meant by this?

6. Introduction. “In addition, we confirmed that hepatocytes of WMP-treated liver grafts retain their function based on ultrastructural observations by scanning electron microscopy after osmium-maceration”

I am not sure how well you can infer that “liver grafts retain their function” from scanning EM

7. Introduction “One of the important physiological functions of hepatocytes is the production and secretion of bile [62], and detailed studies by transmission electron microscopy (TEM) on the capillaries involved in bile are performed [63].”

The second half of the sentence does not appear to follow the first. What is meant here?

8. Materials and methods / Animals/ “cross-bred Large-Yorkshire, Landrace, and Duroc pigs”

Large-Yorkshire pigs. I believe the breed can either be called a Yorkshire pig or a Large White pig or a Large White Yorkshire pig.

9. Materials and methods / Machine perfusion preservation.

The authors have developed a machine that supplies the PV with perfusate that has not passed through an oxygenator, while the HA is supplied by oxygenated perfusate. This is not a feature of any hypothermic oxygenated protocols in current practice – HOPE perfuses the portal vein alone with oxygenated perfusate (and there is no HA perfusion) and D-HOPE perfuses both HA and PV with oxygenated perfusate. There is a limited amount of oxygen that can be dissolved in UW-MPS solution and in the clinical studies investigators have tried to optimize this, as opposed to deliberately dilute it down. At normothermia using a blood based perfusate I could understand the approach they have followed, but it seems illogical at these temperatures with UW-MPS as a perfusate.

10. Materials and methods / Preparation and preservation / “peeling around the hepatic artery and portal vein were performed”

What is meant by “peeling”

11. Materials and methods / Preparation and preservation / “euro-Collins”

Why was Euro-Collins used for liver preservation; it was long since shown to be inadequate as a liver preservation solution. It would have been more appropriate to flush the livers with UW solution.

12. Materials and methods / Preparation and preservation / “University of Wisconsin gluconate”

By this do the authors mean machine perfusion UW solution, or was it simple cold storage UW to which they added gluconate, in which case how much was added?

13. Materials and methods / Preparation and preservation / “The degree of bile duct dysfunction at four hours after MP…”

ALP will tell about cellular release of the enzyme, but not about bile duct function or dysfunction. I realize this is sematic, but it is important to appreciate what you are actually measuring; some readers may not be so discerning

14. Results. General comment

The results are studies of the liver biopsies after different periods of MP, with no reperfusion step. It would be much more interesting and clinically meaningful to see what the effects of the different perfusion interventions are post reperfusion.

15. Results. Figures. The figure captions need to be with the figures not in the middle of the text

16. Results. “These findings indicated that, MP preservation, especially WMP, of liver grafts after warm ischemia enabled the ultrastructure of the bile canaliculi to be maintained and restored, although these effects were insufficient for the recovery of bile production.”

In hypothermic or even at 22°C I would not expect bile production, particularly in the absence of a substrate.

17. Discussion. “Because the actual hepatocyte metabolism due to interaction with other organs cannot be reproduced [96], even monitoring of liver grafts during reperfusion using blood from the same individual after perfusion storage does not seem sufficient to evaluate the liver graft function.”

I disagree. Reperfusion at normothermia ex situ would further inform the degree to which the two preservation modalities affect liver histological appearance.

18. I am not a histologist so cannot comment on the changes between the electron micrographs, but recommend an independent review of these

Reviewer #2: Ishihara et al have presented their work on ultrastructural characteristics of bile canaliculus in machine perfused porcine livers following donation after circulatory arrest. It’s an extension to their similar previous study on hepatocyte ultrastructure. The authors retrieved porcine livers after 1 hour of warm ischaemia and subjected grafts for four hours of machine perfusion, either hypothermic at 8 C or rewarmed to 22 C. They conclude that warm preservation is better in preserving and restoring the ultrastructure in bile canaliculus. I congratulate the authors on these important findings. In our experience, we are confident in predicting hepatocyte function in normothermic machine perfusion livers but cholangiopathy is still encountered in such grafts. Assessment of cholangiocyte function is still experimental and needs more investigation. My comments for this manuscript are as follows:

1. Line 55, page 3. Machine perfusion is a more accepted term and is what mentioned in the abstract and used elsewhere in the manuscript. I would suggest changing it to machine perfusion than 'mechanical perfusion'.

2. Line 96, 97, page 4. Flow should be pulsatile for the hepatic artery and non-pulsatile for portal vein. It has been reported the other way around.

3. Line 114, page 5. Induce cardiac arrest in place of induce to cardiac arrest.

4. Bile production, as stated by the authors, was not seen after HMP or WMP. However, bile production is not seen in hypothermic liver perfusion even in clinically transplantable livers. Bile production is not a marker of viability in HMP livers because of low metabolic activity at low perfusion temperature.

5. It would be interesting to find ultrastructural changes in normothermically perfused livers with blood-based perfusate and comparison with HMP and WMP. Also the structure of first and second-order bile ducts. It has been acknowledged by the authors in the discussion section and I would strongly recommend it.

6. Bile pH as a predictor of cholangiopathy needs validation in larger series and has been reported only by groups in studies that included few patients.

7. Lastly, now the wider consensus is to use the term ‘circulatory arrest’.

6. PLOS authors have the option to publish the peer review history of their article (what does this mean?). If published, this will include your full peer review and any attached files.

Reviewer #1: No

Reviewer #2: No

---

## [Author Response · Author response to Decision Letter 0]

11 Apr 2020

We have addressed all of the concerns raised by each of the reviewers.

We express our sincere thanks to anonymous reviewers for their helpful comments on earlier version of this paper.

---

## [Decision Letter · Decision Letter 1]

15 May 2020

The ultrastructural characteristics of bile canaliculus in porcine liver donated after cardiac death and machine perfusion preservation

PONE-D-19-34589R1

Dear Dr. Bochimoto,

We are pleased to inform you that your manuscript has been judged scientifically suitable for publication and will be formally accepted for publication once it complies with all outstanding technical requirements.

With kind regards,

Kourosh Saeb Parsy

Academic Editor

PLOS ONE

Additional Editor Comments (optional):

Reviewers' comments:

Reviewer's Responses to Questions

**Comments to the Author**

1. If the authors have adequately addressed your comments raised in a previous round of review and you feel that this manuscript is now acceptable for publication, you may indicate that here to bypass the “Comments to the Author” section, enter your conflict of interest statement in the “Confidential to Editor” section, and submit your "Accept" recommendation.

Reviewer #2: All comments have been addressed

2. Is the manuscript technically sound, and do the data support the conclusions?

Reviewer #2: Yes

3. Has the statistical analysis been performed appropriately and rigorously? 

Reviewer #2: N/A

4. Have the authors made all data underlying the findings in their manuscript fully available?

Reviewer #2: Yes

5. Is the manuscript presented in an intelligible fashion and written in standard English?

Reviewer #2: Yes

6. Review Comments to the Author

Reviewer #2: (No Response)

7. PLOS authors have the option to publish the peer review history of their article (what does this mean?). If published, this will include your full peer review and any attached files.

Reviewer #2: No

---

## [Editor Report · Acceptance letter]

20 May 2020

PONE-D-19-34589R1 

The ultrastructural characteristics of bile canaliculus in porcine liver donated after cardiac death and machine perfusion preservation 

Dear Dr. Bochimoto:

I am pleased to inform you that your manuscript has been deemed suitable for publication in PLOS ONE. Congratulations! Your manuscript is now with our production department. 

With kind regards,

on behalf of

Dr. Kourosh Saeb Parsy 

Academic Editor

PLOS ONE